# Evaluating the Impacts of Autonomous Vehicles' Market Penetration on a Complex Urban Freeway during Autonomous Vehicles' Transition Period

Mohammad A. R. Abdeen [1,*], Ansar Yasar [2], Mohamed Benaida [1], Tarek Sheltami [3]⬤, Dimitrios Zavantis [2]⬤ and Youssef El-Hansali [2]

1   Department of Computers and Information Systems, The Islamic University of Madinah,
    Medina 42351, Saudi Arabia
2   Transportation Research Institute (IMOB), Hasselt University, 3500 Hasselt, Belgium
3   Computer Engineering Department, Interdisciplinary Research Center of Smart Mobility and Logistics,
    King Fahd University of Petroleum and Minerals, Dhahran 31261, Saudi Arabia
*   Correspondence: mohammad.abdeen@iu.edu.sa

**Abstract:** Autonomous vehicles (AVs) have been a rapidly emerging phenomenon in recent years, with some automated features already available in vehicles. AVs are expected to potentially revolutionize the existing inefficient state of urban transportation and be a step closer to environmental sustainability. This study focuses on simulation modeling in assessing the potential effects of autonomous vehicles (AVs) and on mobility and safety by developing a framework model based on traffic microsimulation for a real network located in Al-Madinah, Saudi Arabia. The market penetration rates (MPRs) will not reach 100% in the near future; instead, penetration will progressively increase. As a result, in our study, we investigated the potential effect of AV technology in five different AV market penetration rates: 0% (baseline), 25%, 50%, 75%, and 100%. The results suggest that Avs significantly improve the network's safety and operational performance at high penetration rates. Specifically, estimated vehicle delays decreased by 26%, 34.4%, 63.7%, and 74.2% for 25%, 50%, 75%, and 100% AV penetration rates, respectively. Finally, we think this study will help decisionmakers over in the long-term in their attempts to achieve sustainable development through the optimal integration of innovative and novel technologies.

**Keywords:** traffic microsimulation; road safety; Vissim; driving behaviors

## 1. Introduction

Traffic congestion, delays, expenses, and lost productivity pose concerns in most countries with major overcrowded cities [1]. Billions of USD have been invested in some national road networks over the previous several decades to reduce fatalities, traffic congestion, and vehicle-related injuries caused by human error [2].

As a practical alternative, modern research supports the employment of innovative wireless communication in conjunction with autonomous and connected vehicles [3–5]. Ramp meters, dynamic signal timing, and shifting speed restrictions are just a few of the automation solutions that have been proposed to help relieve traffic congestion [3]. In the form of connected and automated vehicles (CAVs), improved processing power and sensors as well as communication and connection capabilities are making their way into the automobile sector, promising to transform the nature of transportation and how people and products are moved from one location to another [4–6].

Recently, automated vehicle (AV) applications have attracted much attention regarding mobility and safety performance of future road transport systems. The autonomous vehicle has the potential to revolutionize the existing inefficient state of urban transportation, offering major mobility benefits while bringing us a step closer to environmental

sustainability [7–9]. It is estimated that connected and automated vehicles can reduce traffic-related fatalities by 30,000 each year in the United States alone [10]. According to the Insurance Institute for Highway Safety [11], nearly one out of every three fatal crashes could be avoided if the first level of autonomous vehicles' crash-avoidance technologies were simply used. Excessive speeding, driving too close to the next car, and other high-risk driving practices might increase the probability of an accident. Based on the proportion of collisions in which human error acts as a contributing component and the premise that because human factors are shown to influence 94% of all crashes [12], this factor will not exist in autonomous road conditions, it is estimated that introducing CAVs will result in a 90% reduction in crash rates [13].

Automated vehicles (AVs) are a rapidly evolving technology that has recently been used to develop and deploy fully connected transportation systems [14–16]. Some automated technologies, such as self-parking, automated braking, lane departure warning, and blind spot monitoring, are already available in vehicles [13]; Nissan, Volvo, Google, Apple, and Uber expected to offer commercially available automated-driving technology in a variety of automobile models in 2020 [13]. Self-driving vehicles were predicted to be available by 2020, followed by urban autopilot mode by 2022, and fully driverless cars will appear on a massive scale no earlier than 2025 [13]. Since 2014 and 2016, AV testing on highways has been authorized in four states in the United States and Australia. The market penetration rate of AVs is estimated to reach between 24% and 87% by 2045 [17,18].

Several studies [16,19,20] have investigated the potential benefits of AVs and found that they can reduce collisions, ease traffic, reduce fuel consumption, and provide flexibility to people who do not have access to transportation, such as the elderly and people with disabilities [21,22]. A number of studies [19,23] have investigated the influence of changes in AV penetration on traffic flow. Zhang et al. [23] found that adopting AVs at a low market penetration rate (MPR) of 2% can reduce 90% of the parking demand for clients. Moreover, Yang et al. [19] observed that under high-volume conditions, when the MPR hits 25%, the likelihood of secondary crashes can decrease by up to 33%. Furthermore, at low MPR levels of roughly 5%, the probability of secondary collisions can be lowered by about 10% if traffic volumes are high.

Moreover, a number of studies have explored the performance of traffic control measures (e.g., VSL) and roadway capacity at various AV penetration levels. Perraki et al. examined the VSL and ramp-metering strategies in 20%, 50%, and 100% of AV cases [24]. Kan et al. used several penetration rates to analyze the effect of CAVs on roadway capacity. They found that the road capacity did not improve much with penetration rates lower than 60%, but 100% penetration increased the capacity by 90% [25].

According to some authors [21,26–28], connected and autonomous vehicles (CAVs) can prevent conflicts and decrease fatal accidents by at least 40%. Moreover, CAVs' driving systems are said to be able to adapt as they approach hazardous locations, potentially reducing delays and increasing safety while navigating busy traffic crossings [21,29,30]. All these factors have the potential to alter transportation systems drastically.

Many of the potential impacts of AVs identified in the research remain speculative in nature due to the limited availability of fully automated technology on the market. However, with the advancement of simulation tools, opportunities are becoming available to explore the effects of automated vehicles, or various aspects of automation, well-before their widespread deployment in the transportation system [21,31]. For example, traffic simulation models allow researchers to investigate the traffic performance of a road network, such as density, overall speed, delays, and potential congestion. In addition, technologies that are currently unavailable or only partially available, such as car platooning, can be tested in a safe simulation environment [32,33].

AV simulation research aims to produce detailed simulations by developing an integrated multi-level simulation platform that includes traffic, sensor (robotics), and network simulators [34]. Vissim, SUMO, and Mas T2erLab have facilitated traffic simulation, while USARSIM, MDDVS PreScan has been used for robotics simulation and NS2 and NS3 have

enabled network simulations that can accurately imitate CAVs [34]. Vissim has been used widely to simulate AVs in a number of investigations.

The vast majority of automobile manufacturers are working on autonomous vehicles, and field testing is taking place on highways and in cities. These tests showed that AVs will pose a wide range of new issues and transform existing road networks in various ways. Certain difficulties, such as whether existing highway and urban infrastructure can accept AVs, are still being investigated. Furthermore, the inherent issues stemming from the interplay between human-driven vehicles and AVs during the transition period remain mostly unknown [34].

Because AVs are not currently present in traffic streams, it is difficult to forecast the potential effects of this disruptive paradigm change on existing transportation systems. Therefore, before implementing AVs on roadways, it is vital to utilize and evaluate them in controlled contexts. To this end, significant efforts must evaluate AVs in a wide variety of unique situations, leveraging research undertaken regionally and worldwide.

Many countries are now racing to create a transportation environment that is more future-oriented and user-centric. Several years ago, Saudi Arabia recognized the need for more sustainable, robust, and human-centric urban mobility systems. As a result, Saudi Arabia is well-positioned to become a global leader in this industry, according to Saudi Vision 2030, which laid out a bold plan for the future [35].

With the kingdom's strong economic growth and diversification, characterized by a greater focus on non-oil businesses, the nation's transportation sector has shown robust development through increasingly practical autonomous vehicle (AV) plans [35]. Saudi Arabia set aside USD 500 billion to spend in the Neom economic zone, a smart metropolis that will be created from the ground up and whose residents will use completely autonomous vehicles (AVs) to move around. Further, the Kingdom of Saudi Arabia's Ministry of Transport and Logistics Services inked a memorandum of understanding (MoU) with Navya to promote the deployment of autonomous vehicles in the kingdom [35]. Saudi Arabia claims this will assist the country in becoming one of the world's first markets to construct large-scale AV infrastructure [35].

As part of the nation's efforts to achieve this goal, this study serves as part of a project in Al-Madinah City, Saudi Arabia, that aims to advance transportation systems by establishing pilot settings for novel sensing technologies while considering the coexistence of base case autonomous cars.

The main objective of this paper is therefore to investigate the potential effects of autonomous vehicles (AVs) on traffic flow and delays. Vissim was chosen as a traffic microsimulation tool to represent human-driven cars (base case vehicles) and autonomous vehicles (AVs). Shorter headways and more aggressive acceleration were specifically considered more assertive AV characteristics. The case study compared conventional and autonomous vehicles (AVs) at a road section in Al-Madinah City, Saudi Arabia. While market penetration rates (MPRs) will not reach 100% in the near future, they will progressively increase. As a result, it is imperative to investigate the safety benefits of AV technology in various MPR scenarios.

This study aims to help decisionmakers achieve sustainable development in the long-term. Although detailed investigations may have been conducted in other nations, such efforts have not yet sufficiently occurred in the context of Al-Madinah City, to our knowledge. The remainder of this paper is organized as follows: The methodology is outlined in the next section, followed by key results and discussion in Section 3. The paper concludes with a summary of key findings and directions for future research.

## 2. Methodology

We opted to use Vissim for this study because it appeared the most viable and effective traffic micro-simulation package with which to evaluate the potential influence of AVs on the safety and operational performance of the traffic stream on a section of highway [33]. Vissim includes different pre-defined modules that can be programmed to simulate various

technologies based on the behavior of vehicles. By employing Wiedemann's psycho-physical car-following logic, it models traffic on urban streets (the Wiedemann 74 model), highways, and freeways (the Wiedemann 99 model). The Vissim 11 software therefore served as the traffic microsimulation platform for this study. In a simplistic approach, typical driving behaviors of autonomous vehicles (AVs) can be described based on some basic AV characteristics. First, AVs require less lateral space. Thus, road lanes can be narrower, and consequently, the number of lanes can be increased without increasing the road width. Furthermore, AVs maintain a smaller standstill distance and shorter headway, react to signals without reaction time, strictly maintain the desired speed (without distribution and oscillation), and accelerate and decelerate without distribution. AVs are represented by a new vehicle type, "Car-AV," which is included in the new vehicle class "Car-AV" [33].

Our study considered five basic scenarios. The first involves simulating and testing the efficiency of the highway section with only conventional vehicles in the traffic stream. The second scenario involves combinations of conventional vehicles and AVs with different market penetration rates, 25%, 50%, and 75%, and the last scenario simulates the operational efficiency of the segment when only AVs are on the road (100% MPR). The base scenario was designed before creating the other scenarios. Essentially, the only difference in the modification of each scenario is the driving behavior involved.

A three-lane motorway section (in each direction) in Al-Madinah, Saudi Arabia served as a test environment. Conventional vehicles consisting of 98% base case cars with a common desired speed distribution (90–100 km/h) and 2% of trucks with a standard desired speed distribution (80–90 km/h), and AVs with a constant desired speed (80 km/h), were utilized for each MPR case (PTV, 2021). The overall architecture of the proposed simulation framework is presented in Figure 1.

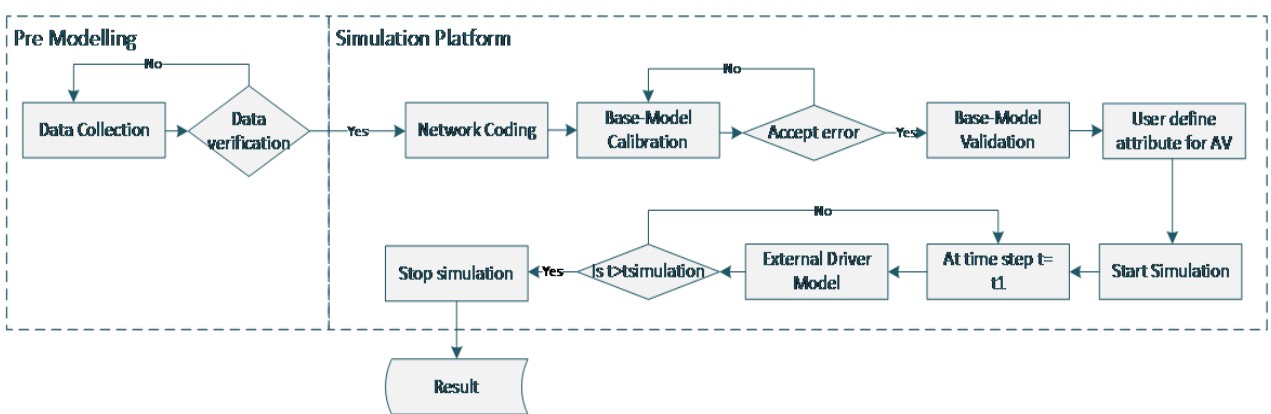

**Figure 1.** Simulation framework for AV evaluation process.

### 2.1. Simulation Platform

Vehicular movement in Vissim is determined by other vehicles' movement and surrounding infrastructure, and the driver and vehicle are treated as a single entity. One of the four driving modes, free driving, approaching, braking, and following, is assumed. In this paper, we used the Wiedemann 74 and 99 models to simulate the human driving behavior, as recommended by Vissim for urban and interurban traffic, since they contained more adjustable driving behavior parameters that could contribute to a more accurate safety-oriented calibration of the baseline model.

#### 2.1.1. Test Base Model Calibration and Validation

Initially, a scaled base model using the Vissim interface by utilizing Google Earth enabled both directions of the road segment, lane width, length of links, and merging and diverging areas to be drawn. The merging and diverging areas were designed following guidelines from literature focusing on motorway merging areas, and the model designed did not contain any intersections. The test base was calibrated and validated on the basis

of local driving-behavior parameters from the Wiedemann 74 car-following models to ensure that it represented the actual case conditions, using the total flow and average speed for calibration and the travel time for validation [32,36]. The Geoffrey E. Havers (GEH) statistical procedure was used to evaluate the calibration results [32,36]. The GEH statistics were calculated using Equation (1):

$$GEH\ Statistic = \sqrt{\frac{(0-M)2}{0.5(0+M)}} \tag{1}$$

where *O* is the observed traffic value and *M* is the measured traffic value. The calibrated model was then validated using the average speed. Two more statistical analyses (goodness-of-fit measurements) were conducted to validate the calibrated model: the root means square error (RMSE) and the coefficient of correlation (CC) [32,36]. The RMSE measures the percentage deviation of the simulation output from observed data. Equation (2) is used to calculate RMSE value:

$$Root\ Mean\ Square\ Error\ (RMSE) = \sqrt{\frac{1}{N}\sum_{I=1}^{n}\left(\frac{O-M}{O}\right)^2} \tag{2}$$

The *coefficient of correlation* (CC) indicates the degree of linear association between simulated and observed data. Equation (3) is used to calculate the CC value:

$$Coefficient\ of\ correlation\ (CC) = \frac{1}{n-1}\sum_{i=1}^{n}\frac{\left(y_{sim}-\underline{y}_{sim}\right)-\left(y_{obs}-\underline{y}_{obs}\right)}{S_{sim}S_{obs}} \tag{3}$$

where *n* is the total number of traffic measurement observations; $y_{sim}$ and $y_{obs}$ are means of the simulated and observed measurements, respectively; and $S_{sim}$ and $S_{obs}$ are the standard deviations of the simulated and observed measurements, respectively. An RMSE of less than 15% is considered acceptable for traffic model calibration, and the coefficient of correlation of 85% is deemed acceptable for model calibration [32,36].

2.1.2. Test Base Model Calibration and Validation

In Vissim, AVs can be modeled very easily by modifying some driving behavior parameters according to the assumptions mentioned above and associating this behavior to a dedicated vehicle type and class.

The acceleration or deceleration and acceptable time gap are calculated according to the following assumptions and equations [37,38].

The acceleration or deceleration (*a*) of a vehicle in Vissim during each time step is defined in Equation (4) [34]:

$$a = \frac{\Delta v}{\Delta t} \tag{4}$$

where $\Delta v$ is the difference between current speed and target speed and $\Delta t$ is the time step of the simulation (in this case, 0.1 s). Figure 2 shows the actual and anticipated distance diagram for individual vehicle acceleration and deceleration calculations [34].

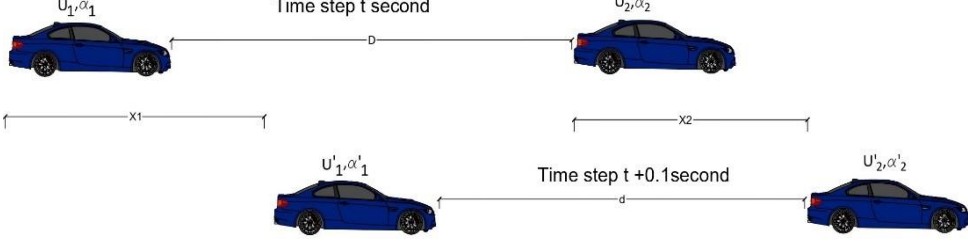

**Figure 2.** Actual and anticipated distance diagram.



For AV simulation (Vissim), the driving behaviors from a compatible dynamic-link library (dll) replaced the internal driving behavior with AV driving behavior and were assigned only to the AV-type vehicles in Vissim. Assuming that the dll-controlled vehicle is not following the preceding vehicle with the desired time gap ($d$), the distance travelled by both cars ($x_1$, $x_2$) and the time gap during time step $t$ and time step $t + 0.1$ can be defined by Equations (5)–(7) [34]:

$$x_1 = u_1 \times t + 0.5\, a_1 \times t^2 \tag{5}$$

$$x_2 = u_2 \times t + 0.5\, a_2 \times t^2 \tag{6}$$

$$d = x_2 + D - x_1 \tag{7}$$

Assuming that the initial speed of the preceding vehicle, and the speed of the vehicle in the back, are not equal. Therefore, the acceleration and timing of the AV needed to achieve the desired time gap are calculated as shown in Equations (8) and (9) [34]:

$$t = \frac{2(x_2 - x_1)}{(u_2 - u_1)} \tag{8}$$

$$a'_1 = \frac{(u_2 - u_1)^2}{2(x_2 - x_1)} \tag{9}$$

The accepted car-following time gap chosen for this study was 0.6 s [37,38].

In our model, base case cars use the default driving behavior "urban" (motorized), while AV cars are controlled by the driving behavior "urban–AV," which was modified from the default behavior in the following way: For the Wiedemann 74 behavior parameters, the average standstill distance for a conventional car is 1.5, while it is 0.75 for an AV; the additive part of safety distance is 2 in a conventional car and 1.5 for an AV; and the multiplicative part of safety distance is 3 for a conventional car and 0 for an AV [33].

AV cars' parameters were modified for the Wiedemann 99 behavior parameters, such as shorter standstill distance (CC0) and shorter safety distance (the lower-headway CC1 and following-variation CC2). Thus, shorter gaps are expected for AVs. Smaller values of the negative following threshold (CC4) and positive following threshold (CC5) reflect AVs' more sensitive reaction to the preceding vehicle's acceleration or deceleration. AVs can strictly follow the desired speed without oscillation, so CC6 is set as zero. Further, AVs can also have more aggressive acceleration (higher CC7 and CC8) and a higher number of observed vehicles due to connected vehicle technology [33]. Although the exact behaviors of AVs remain largely unknown at this stage, the modification of these parameters should reflect the anticipated AV behaviors. Table 1 presents a conventional car and AV parameters set according to PTV's 2020 guide.

**Table 1.** Car-following model parameters for both AVs and conventional vehicles (PTV, 2020).

| Parameter | Description | AV | Conventional Car |
|---|---|---|---|
| CC0: Standstill distance ($m$) | The desired distance between two cars | 0.75 | 1.5 |
| CC1: Headway time ($s$) | The gap in seconds that a vehicle maintains | 0.45 | $0.9 \pm 0.2$ |
| CC2: Following variation ($m$) | The distance in addition to the allowed safety distance that is permissible before the car moves closer to the car in front | 1.5 | 2.0 |
| CC3: Threshold for entering car-following mode | Controls the start of the deceleration process when a driver recognizes a preceding slower car | −8.0 | −8.0 |
| CC4: Negative/positive following threshold | Controls speed differences during car following | ±0.1 | −0.35 |
| CC6: Speed dependency of oscillation | Influence of distance on speed oscillation (the variation in speed around the desired speed) | 0.0 | 11.44 |
| CC7: Oscillation acceleration ($m/s^2$) | The actual acceleration during the oscillation process | 0.25 | 0.25 |
| CC8: Standstill acceleration ($m/s^2$) | Desired acceleration when starting from a standstill | 3.5 | 3.5 |

### 2.1.3. Test Base Model Calibration and Validation

To evaluate the potential influence of AVs in terms of safety and mobility, several measures of effectiveness (MoEs) are used [6,32,36,39]. The safety MoEs include the number of stops, which can serve as an appropriate performance measure representing the stop-and-go shockwaves in the network [14] and average stopped delay, which is the time during which a vehicle is stopped in a queue while waiting to pass through the bottleneck area [6], while the mobility MoEs include bottleneck throughput (capacity), vehicle delay, and travel time. Improving these MoEs indicates the improvement of the operation and safety of traffic flow during the incident. Reducing travel time, reducing delay, and increasing throughput were intended to improve traffic mobility while reducing the number of stops, with the aim of reducing secondary accidents at the incident area and, therefore, improving safety [36].

To measure these metrics from the simulation, we used the built-in evaluation functions in Vissim. The user-defined data collection points were applied to calculate the number of vehicles of particular vehicle classes and the average speeds. Vehicle travel time measurements calculated total travel times and delays ("stops all", "stop delay"), and Vissim computes a vehicle stop when the vehicle changes its speed from any speed greater than 0 to a speed of 0; that is, when the vehicle comes to a standstill [33,36].

## 3. Results Analysis

This section presents the summary results for the comparison scenarios in terms of the overall intersection performance measures of effectiveness (MoEs) in each situation. The base model first was calibrated and then validated manually. The GEH statistic was calculated using Equation (1) and found to be lower than five. All goodness-of-fit measurements (the RMSE and the MANE) fell within an acceptable range of 11% and 13%, respectively.

To evaluate the performance of AV technologies on the arterial section, five MoEs were used (the number of stops, average stopped delay, vehicle capacity, vehicle delay, and travel time). To minimize the effect of the stochastic nature of the Vissim, each scenario was run seven times [36] for 4200 s each, including warm-up and cool-down periods lasting 300 s. To evaluate the operational traffic performance of AVs in terms of capacity, travel time, and travel delay, Figures 3–5 represent vehicle capacity, travel time, and vehicle delay.

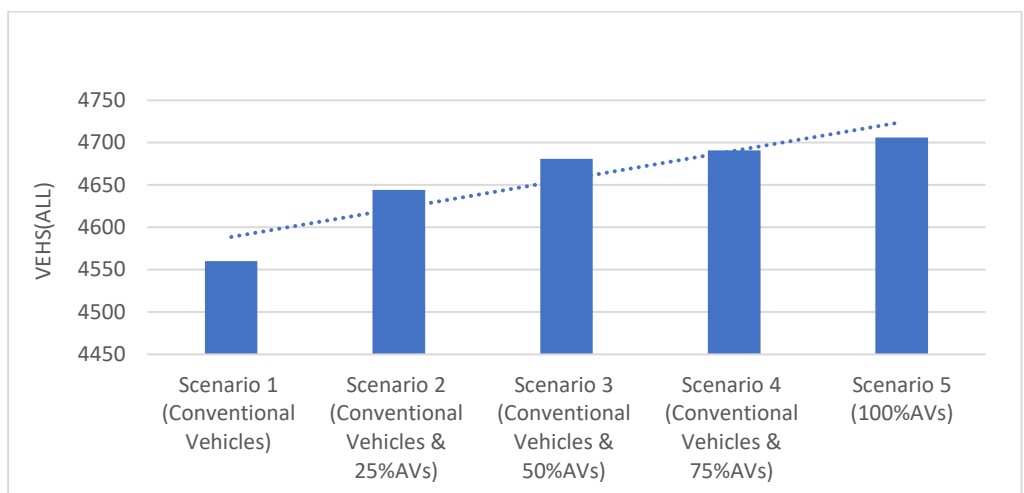

**Figure 3.** Simulation results of average VEHSALL for all scenarios.

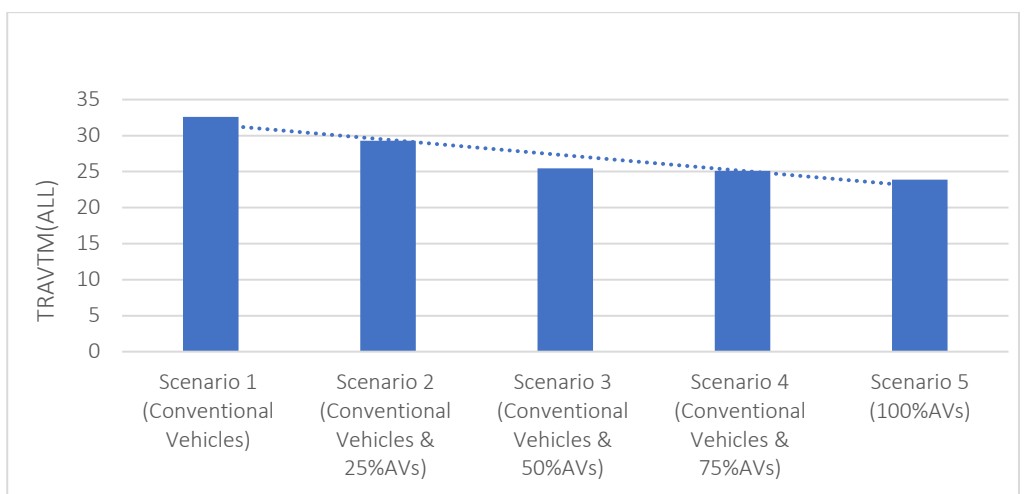

**Figure 4.** Simulation results of average travel time for all scenarios.

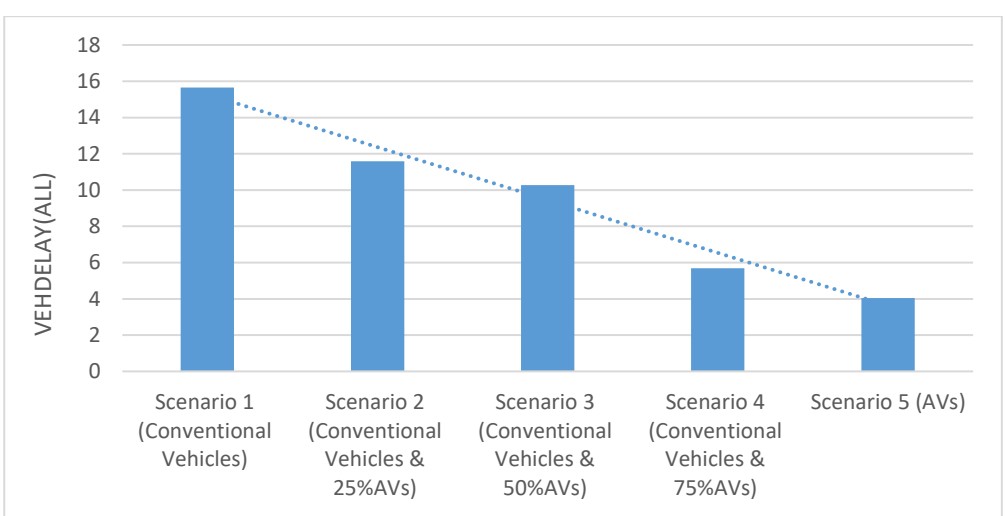

**Figure 5.** Simulation results of average vehicle delay for all scenarios.

Figure 3 shows that the average VEHS capacity of the highway section increased gradually with the increase in AVs' MPR. The capacity increased by 3.2% when the MPR reached 100% (meaning all vehicles on the road were autonomous vehicles), while it was 1.5 when the MPR was 25%.

Similarly, Figure 4 presents results for average travel time. We can see that the average travel time decreased by 26.7% when the traffic stream consisted of 100% AVs on this highway segment.

Results of the average vehicle delay look relevant to the output of the average travel time. Vehicle delays were reduced by 26%, 34.4%, 63.7%, and 74.2%, when the traffic stream consisted of 25% AVs, 50% AVs, 75% AVs, and 100% Avs, respectively.

To evaluate the safety performance of AVs, Figure 6 represents the overall segment of average "stops all" and "stop delay" for all scenarios.

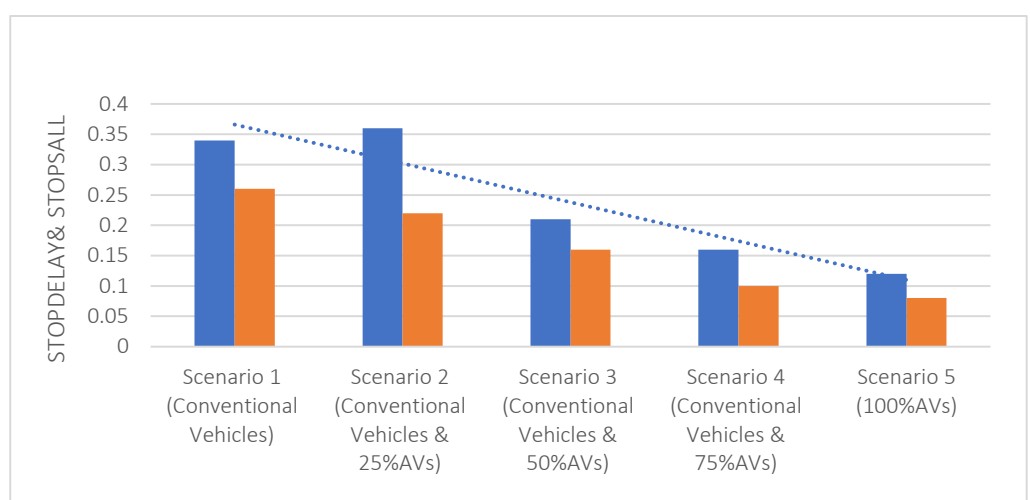

**Figure 6.** Simulation results of average "stops all" and "stop delay" for all scenarios.

Figure 6 shows that the average "stop delay" and "stops all" incidents dropped 64.7% and 69.2%, respectively, in Scenario 5 (in which the highway section traffic was 100% AVs). Table 2 summarizes all the results presented in Figures 3–6. Again, Scenario 1 served as a baseline scenario, to which all other scenarios were compared, and that is why it has a percent change of 0% value for the road section performance MoEs.

**Table 2.** Summary results comparing all segment performances.

| Scenarios | Vehs (All) | Travtm (All) | Stopdelay (All) | Stops (All) | Vehdelay (All) |
|---|---|---|---|---|---|
| Scenario 1 (Conventional Vehicles) | 0% | 0% | 0% | 0% | 0% |
| Scenario 2 (Conventional Vehicles and 25% AVs) | 1.8% | −10.1% | 5.9% | −15.4% | −26% |
| Scenario 3 (Conventional Vehicles and 50% AVs) | 2.7% | −21.9% | −38.2% | −38.5% | −34.4% |
| Scenario 4 (Conventional Vehicles and 75% AVs) | 2.9% | −23.0% | −52.9% | −61.5% | −63.7% |
| Scenario 5 (100% AVs) | 3.2% | −26.7% | −64.7% | −69.2% | −74.2% |

The results in Table 2 show that the use of autonomous vehicles enables promising improvements in traffic safety and efficiency of arterial roads. In summary, based on the results presented in this section, AVs can improve the operational efficiency of urban arterial roads by minimizing travel time and vehicle delays while increasing vehicle capacity. Further, AVs can improve safety performance by reducing "stops all" and "stop delay" incidents.

## 4. Conclusions and Future Research

The most pressing issues that road users face daily include traffic congestion and car crashes. According to previous studies, most traffic accidents are caused by human error. In contrast, the majority of repeated traffic jams are caused by bottlenecks, which arise when traffic demand exceeds available route capacity. Furthermore, higher travel delays and traffic accidents are caused by driver decision making and their unexpected and variable reaction times. As a result, by utilizing new autonomous vehicle technologies (connected and automated), human errors will be reduced, making roadways safer and more efficient by lowering delays and travel time. Automated vehicles (AVs) are projected to move at a constant speed while leaving reduced headways (gaps) between them, improving traffic roadways' operational and safety performance.

The main aim of this study was to evaluate the influence of automated vehicles on traffic performance and safety measures within a roadway segment. The study mimicked the presence of automated vehicles in the network using Vissim microscopic simulation to describe their driving behavior. Vissim allows for assessment of the influence of automated cars with varied penetration rates by distributing vehicle input among existing vehicle types (conventional and automated vehicles), enabling Vissim to serve as an effective evaluation method in this research. Parameters in Vissim such as "look ahead/back distance (maximum and minimum)," "number of observed vehicles," "cooperative lane change," "advanced merging," and "both sides overtaking" represent the performance of ADAS in automated vehicles. In addition, accepting shorter time headways and achieving tighter desired speed distribution close to the speed limit are examples of possible adjusted measures for showing the distinct driving behavior of automated vehicles. Five different AV market penetration rates were tested: 0% (baseline), 25%, 50%, 75%, and 100%.

This study also found that during the transition period (when AVs coexist with conventional vehicles), the roadway segment performs better than when only conventional vehicles are present in the traffic stream. It is projected that AVs will gradually penetrate the car market, eventually phasing out all conventional vehicles, at which point the full benefits of AVs will be realized. The result of these findings shows that traffic congestion on roadways will be lessened as the market for autonomous vehicles expands.

We advocate for generating more scenarios that include a mix of conventional and autonomous vehicles and a wide range of driving behaviors for future work. More realistic scenarios should be assessed under various traffic conditions. Further, more simulation studies on both urban and rural roads with varying traffic conditions are required.

Although environmental factors for automated vehicles, such as $CO_2$ emissions and fuel consumption, have not yet been evaluated in this research, we are planning to assess the environmental effects of CV, and as a result, the economic cost savings, fuel cost savings, and pollution rate reduction.

Making driving easier by reducing congestion and travel times will facilitate sustainable mobility transitions. Various legal, social, economic, and other issues are involved in the transition of the conventional traffic system to the proposed one. These issues will be explored in terms of social acceptance of the system and investments required for implementation in our future research.

**Author Contributions:** Conceptualization, M.A.R.A.; Data curation, M.A.R.A.; Formal analysis, M.B.; Investigation, Y.E.-H.; Methodology, A.Y. and Y.E.-H.; Resources, T.S.; Software, M.B. and T.S.; Supervision, A.Y.; Visualization, A.Y.; Writing—original draft, D.Z.; Writing—review & editing, D.Z. All authors have read and agreed to the published version of the manuscript.

**Funding:** This research received no external funding.

**Institutional Review Board Statement:** Not Applicable.

**Informed Consent Statement:** Informed consent was obtained from all subjects involved in the study.

**Data Availability Statement:** Not Applicable.

**Conflicts of Interest:** The authors declare no conflict of interest.

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
