# Peer review of "Evaluating the Impacts of Autonomous Vehicles’ Market Penetration on a Complex Urban Freeway during Autonomous Vehicles’ Transition Period"

_sustainability, doi:10.3390/su141610094_

Round 1

Reviewer 1 Report

Dear Authors, 
I am very sorry to write this, but in my opinion, the study you have done does not make much sense. Choosing the PTV Vissim tool, which is impossible to simulate road accidents or collisions, is not a good solution. Especially since most road collisions are followed by a stop of two or more vehicles that were involved in the crash. Usually, the road lane is also closed due to abandoned vehicles. Even if we assume very restrictive boundary conditions in the behaviour of drivers focused on aggressive driving, it will still not be adequate for any comparisons because this collision does not occur physically. Your research shows that the more non-aggressive vehicles we add to the road network, the better conditions we will get, so it's obvious. Whether they will be vehicles or autonomous vehicles does not matter much, since we are examining this percentage of vehicles that move in a given way (more or less aggressively). I understand that modelling autonomous vehicles are difficult for this type of research because there are currently no confirmed examples to calibrate such a model on a larger scale, however, in my opinion, PTV Vissim will not give us any meaningful results for the test proposed by you.

Author Response

Please see attched file.

Reviewer 2 Report

 This study attempts to assess the potential impacts of Autonomous vehicles (Avs) on mobility and safety by developing a framework model based on traffic microsimulation for a real network located in Al- Madinah, Saudi Arabia. As the market penetration rates (MPRs) of AVs will increase progressively in the future, the present study investigated the potential impacts of AVs technology in five MPRs, i.e. 0% (baseline), 25%, 50%, 75% and 100%. Results showed that AVs’ introduction improve the network's safety and operational performance as their penetration rates increase.

Some general comments

It is an interesting paper with main objective, according to authors, the contribution of new technologies in transport and in specific of AVs towards sustainability. At this point I would like to recommend to the authors that they should highlight in their text (abstract, conclusions etc. ) that the findings of this research constitute only a small part of the puzzle used by decision-makers in their attempts to achieve sustainable development through the use of AVs. Quantitative models consider a limited parameter set and the choice of parameters is highly subjective. Therefore i believe that although this study is focused on the results of a microsimulation model describing the driving behavior of AVs the authors should also mention that the considered technological shift could also have some not beneficial impacts such as increase in empty vehicles, longer distances traveled  (urban sprawl) etc. It should be mentioned that future research should take into account the interplay of all the driving forces which are integrated in the technology, i.e. policies, operation and planning.

Specific comments

First, referring to the title: is the Al-Madinah network an urban or an interurban network as the chosen driving models, the Wiedemann 74 and 99 by the authors in order to simulate the human driving behavior were recommended by VISSIM for interurban traffic (lines 178 & 179). In addition it is referred as a three-lane motorway section (Al Madinah, Saudi Aribia). That means that there are three-lanes per direction or in total?  

A great weakness of the paper is the not correct use of acronyms and symbols. In specific either they are not kept constant through the text or they are not followed by the full words. Some examples are given below:

Line 198:  Formula 2 is used to calculate MAPE value.  What the acronym MAPE stands for, when formula 2 refers to Root Mean Square Error (RMSE)?

Also in line 205: RMSNE of less than 15%. Is RMSNED written instead RMSE?

Line 199: A coefficient of correlation is given as, (CC), when later in line 205 is given as A.

Line 203: Vissim and sobs instead of sims and sobs.

Line 220: What is dll controlled vehicle?

In equation 7 what is D for?

Line 247: What MOHs means? Probably it is written instead MOEs (Measurements of Effectiveness).

In line 169 what is the meaning of the phrase ‘and one consisting of autonomous cars with a constant desired speed (80 km/h) ‘when in the base case there are no AVs?

In line 175 the phrase “This movement of other vehicles and the surrounding infrastructure determine vehicle movement in VISSIM" is not clear.

Another point that it should be clarified is what is the meaning of the values for Car following model parameters for both AVs and conventional vehicles given in paragraph from line 233-243 , when there are different values for these parameters in Table1?

Another weak point is  the presentation of figures 3-6. The absence of parameters in the y-axes makes difficult the interpretation of the results. 

Finally, concerning the references these are not written with a constant format.

Author Response

Please see attched file.

Reviewer 3 Report

The article is sound and on an acute topic. However, there are several simulations of AV impacts in the literature; some of them using VISSIM, too. The authors have referenced some of them. But they do not compare/benchmark their work and results to them. This is a shortcoming that needs improvement. Also, the study objectives and hypotheses need to be more clearly presented, better in a separated sub-section. Finally, conclusions' section is too short and not very clear. Key findings would be better to be presented in bullet format, followed by future work steps. 

Author Response

Please see attched file.

Reviewer 4 Report

line 59-61: Here you mention that several companies plan to have commercial AVs by 2020... however it is 2022 and none of them do. Be careful in what sources you are citing...

line 67-68: Not sure what the relevance to your paper is in mentioning the demand for parking in Zhang et al.

line 77-79: Duplicate sentence

line 246-256: You need more here to explain why stops is being used as a proxy for safety. This connection is not intuitive and in the results simply showing that there are fewer stops does not equate increased roadway safety.

Figures 3-5: what is the Y axis? please add label. I assume in figures 4 and 5 this is minutes, but still, a label would be helpful since that time could also be something like hours per week or something... 

Overall a good paper. Its well written and the methodology seems sound, although I do not do simulation like this. With that being said, there are some issues that could be raised in the discussion which I think would add significantly to the paper. 

First, is the issue of the transition period. I've read other studies that seem to indicate that this period may be more dangerous due to human driver uncertainty regarding how AVs operate. Likewise, AVs may be less efficient simply because they cannot adapt and respond so well to higher numbers of human drivers. Since your paper looks at this transition at different AV penetration rates, it seems that you could say something more about this in your paper, since your own simulation assumes a steady decrease in congestion and delays and steady increase in safety.

Second, you mention initially in your introduction that infrastructure is likely to play a role in AV market penetration, so it could be useful to address this more directly in your analysis. Do your different penetration rates also assume some level of new infrastructure and how would investment in that infrastructure limit the market penetration rates?

Lastly, this is more of a general comment to you as researchers than as something that should be addressed in the paper... but it is more about your overall approach to the topic of AVs. At the end of the day, making driving easier by reducing congestion and travel times, is counter-productive for sustainable mobility transitions. You mention at the start of your paper that AVs will likely increase transport sustainability, but don't go on to mention how exactly this would happen. I would push you to reconsider this position in its entirety and instead I would argue that AVs as deployed in a way that makes driving private or semi-private vehicles faster, safer (and likely less expensive) is entirely opposite the goals cities need to pursue to make transport more sustainable and to promote more sustainable urban development patterns. At the end of the day, cars require a lot of space, regardless of whether they are self-driving or not. Simply increasing the throughput of a highway for cars isn't necessarily the most efficient way to move people... We know this and research like yours continuously puts AVs in a much more positive light without critically thinking about any of the possible negative impacts.

Again, this isn't something to address in the paper, but something to keep in mind as your conduct future research. For the paper itself, overall good, just needs a few minor revisions.

Round 2

Reviewer 1 Report

The authors answered all questions and doubts.  

Author Response

No remarks found in review round 2 so synthetic paper changes done

Reviewer 2 Report

I am satisfied with the authors' responses to my comments. Therefore i agree with the publication of the article in the Journal of Sustainability.

Author Response

(The authors gave the same response as above.)
